# Temporal Visiting-Monitoring Feature Interaction Learning for Modelling Structured Electronic Health Records

## Abstract

Electronic health records (EHRs) contain patients' longitudinal visit records, and modelling EHRs can be applied to various clinical prediction tasks. Previous works primarily focus on visit sequences and perform feature interaction on visit-level data to capture patient states. Nonetheless, incorporating finer-grained monitoring sequences simultaneously in structured EHRs, where each visit involves multiple monitoring sessions, can improve prediction performance. However, these studies have not accounted for the relationships between visit-level and monitoring-level data. To fill this gap, we propose an EHRs modelling method aimed at modelling the dynamic interaction between visit-level and monitoring-level data and capturing finer-grained health trends. We first capture the dynamic influence between medical data, and then perform a visiting-monitoring feature interaction on the relationships between visit data and monitoring data, to obtain the representation of patients' state for clinical prediction. We conducted extensive experiments on disease prediction and drug recommendation tasks, with MIMIC-III and MIMIC-IV datasets, demonstrating that our method outperforms state-of-the-art models significantly.

## 1 Introduction

Electronic health records (EHRs) contain sequential visit records, including information such as diagnoses and prescriptions (Johnson et al., 2016; Pollard et al., 2018; Johnson et al., 2023). Various clinical prediction tasks based on EHRs have been conducted, such as disease prediction (Choi et al., 2016; Ma et al., 2020a; Chen et al., 2024), drug recommendation (Zheng et al., 2021; Yang et al., 2023b), and mortality prediction (Choi et al., 2017; Gao et al., 2020; Zhang et al., 2021). Modelling EHRs offers a comprehensive, real-time analysis of patients and supports quick and accurate clinical decision-making. Previous works have mainly focused on learning patient health trends from visit sequences, but recent research (Bhoi et al., 2024) shows that incorporating monitoring sequences from structured EHRs captures finer-grained health trends, improving prediction performance. As shown in the left part of Figure 1, structured EHRs contain two levels of medical events: (1) visit-level events, such as diseases, procedures, and drugs, and (2) monitoring-level events, such as lab test results reflecting the patient's health state, where a single visit can involve multiple monitoring sessions, such as those in intensive care unit (ICU) settings.

How to model the complex relationships between medical events for feature interaction learning has become a major challenge in EHRs modelling. The first type of work (Poulain & Beheshti, 2024; Li et al., 2024), as shown in Figure 2(a), analyzes correlations and constructs relationships between events within the same visit, but the relationships across time points are relatively weak. The second type of work (Jiang et al., 2023; Chen et al., 2024), as shown in Figure 2(b), builds pathways based on event recurrence across visits but does not fully account for the finer-grained monitoring sequences, making it challenging to capture finer-grained patient health trends. Recent research (Bhoi et al., 2024) incorporates finer-grained monitoring sequences from structured EHRs, as shown in Figure 2(c). This suggests applying a similar temporal modelling method to monitoring sequences as used for visit sequences.

Figure 1: (1) Left: In structured EHRs data, not only does a single patient have multiple visits, but each visit also includes multiple monitoring sessions. (2) Right: Dynamic pathological relationship between visit-level events and monitoring-level events.

However, a limitation is that it does not consider the relationships between the visit and monitoring sequences. As illustrated in the right part of Figure 1, in real-world clinical scenarios, there is often a dynamic pathological relationship between monitoring events and visit events. For example, hypertension can cause elevated blood pressure (detected by lab tests). When blood pressure is high, patients may need to take blood pressure drugs to lower it. As the blood pressure decreases, the symptoms of hypertension are alleviated. This pathological relationship reflects the interplay between visits and monitoring events and captures fine-grained patient health trends. However, existing methods fail to model these relationships in structured EHRs, resulting in sub-optimal performance.

To fill the aforementioned gap, as shown in Figure 2(d), we propose a temporal cross-level (visiting-monitoring) feature interaction learning method to model the dynamic pathological relationships between visit and monitoring sequences for EHRs modelling, named CrossMed. Specifically, we first estimate the influence between monitoring and visit events, then construct a temporal cross-level interaction graph, creating a sub-graph for each monitoring session. Within each sub-graph, we model the influence of monitoring on visit events, and for consecutive sub-graphs, we model the response of visit events to the next monitoring step. We then perform feature interaction learning, updating event representations along the graph. Finally, we aggregate event representations into patient representations for clinical prediction. To summarize, we make the following contributions:

- To the best of our knowledge, we are the first to model the pathological relationships between visit events and monitoring events in structured EHRs.

- We propose a temporal visiting-monitoring feature interaction learning method based on the pathological relationship between visit event and monitoring event, to capture finer-grained patient health trends.

- We conducted extensive experiments on two real-world medical datasets, including both disease prediction and drug recommendation tasks, to demonstrate the superior performance of our method compared to baselines.

## 2 PRELIMINARIES

### 2.1 PROBLEM DEFINITION

**Data Format.** Structured EHRs contain multi-level continuous clinical records of patients. In the record, each patient is represented as $H = \{V_1, V_2, \ldots, V_T\}$, where $V_t$ denotes the $t$-th clinical visit of the patient for $t \in [1, T]$. For each clinical visit $V_t$, we have $V_t = \{D_t, P_t, R_t, M_t\}$, where $D_t$, $P_t$, $R_t$, $M_t$ represent the diseases, procedures, drugs, and monitoring information of the patient, respectively. Specifically, for diseases $D_t$, a patient's single visit $V_t$ may be associated with multiple diseases simultaneously, hence we adopt multi-hot encoding to denote the information of the disease $D_t \in \{0, 1\}^{|D|}$ with $|D|$ as the total number of disease types. Both procedures[1] $P_t \in \{0, 1\}^{|P|}$ and drugs $R_t \in \{0, 1\}^{|R|}$ similarly employ the multi-hot encoding with $|P|$ and $|R|$ as the total number of procedure types and drug types, allowing patients to have multiple procedures

---

[1]Procedure is mostly recorded as the surgery type.

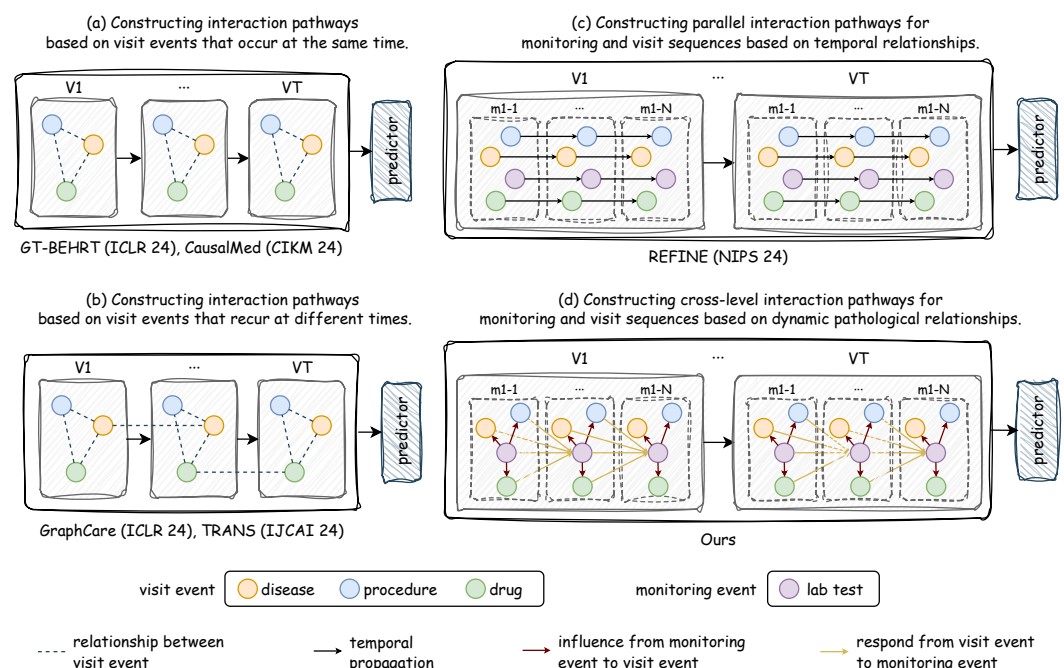

Figure 2: Different feature interaction learning methods in modelling EHRs for clinical prediction.

performed and be recommended with multiple drugs in a single visit $V_t$. In addition, the monitoring information $M_t$ is a finer-grained sequence that represents continuous changes in the patient's health state reflected by monitoring events (*e.g.*, lab test result) during the $V_t$. It is represented as $M_t = \{m_{t,1}, m_{t,2}, \ldots, m_{t,N}\}$, where $m_{t,n} \in [0,1]^{|M|}$ is a normalized vector denoting the health state of $n$-th monitoring session at visit $V_t$, for $n \in [1, N]$, and $|M|$ refers to the total number of categories for all monitoring events.

**Task1: Disease Prediction.** Given the patient health record $H$, disease prediction aims to learn a function $f_{DP}(\cdot)$ that predicts the disease $D_t$ at the end of the visit sequence.

**Task2: Drug Recommendation.** Given the patient health record $H$, drug recommendation aims to learn a function $f_{DR}(\cdot)$ that recommends drugs $R_t$ at the end of the visit sequence.

In this sense, these two tasks can be regarded as multi-label classification problems.

## 2.2 RELATED WORKS

**EHRs modelling in clinical prediction.** In recent years, researchers have increasingly used data mining to develop EHRs modelling in clinical prediction. (1) The first type of research (Choi et al., 2016; Jin et al., 2018; Liang et al., 2021; Wu et al., 2022; Waghmare et al., 2024) focuses on patient state and employs various methods such as attention models, LSTM networks, and Markov decision processes for clinical prediction. However, these approaches often overlook the interactions between medical events. (2) The second type of research focuses on the relationships between multiple medical events, using relational networks to enhance feature interaction. Techniques such as structure learning (Zheng et al., 2021; 2023), causal discovery (Sun et al., 2022b; Li et al., 2024), and bias reduction (Zhao et al., 2024) are used to strengthen the relationships between medical events in graph networks. However, these methods often rely on generated relationships, lacking clear medical significance and sufficient granularity. (3) The third type of research enhances patient representation by integrating domain-specific knowledge. Yang et al. (2021b; 2023b); Chen et al. (2023) leverage molecular data, while Choi et al. (2017); Ma et al. (2018); Shang et al. (2019a) use medical ontologies. Bhoi et al. (2024) combines lab tests with drug-drug interaction databases. However, these methods are limited by their heavy reliance on external knowledge. Our method belongs to the sec-

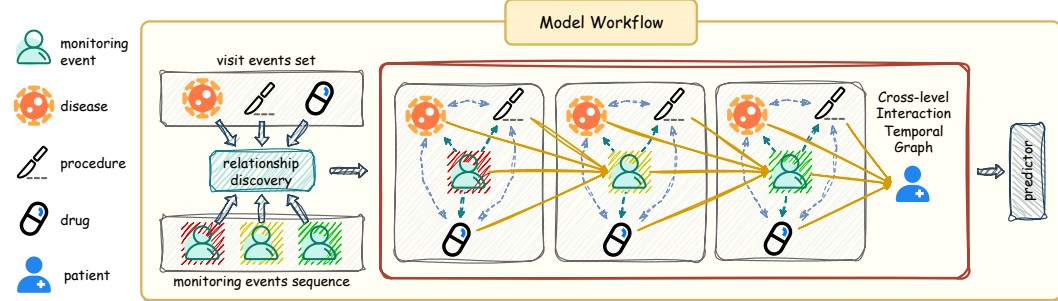

Figure 3: CrossMed consists of relationship discovery, graph construction, and feature interaction. (1) Starting from the workflow's left side, it models relationship weights between different levels of medical events in the relationship discovery stage. (2) Next, as shown in the red box on the right, it constructs graphs based on event types and time. (3) Finally, it performs feature interaction to integrate the heterogeneous relationships into patient representations, which are used for clinical prediction tasks. Relevant legends are displayed on the left side of the workflow.

ond category mentioned above, driven by the latent pathological relationship between monitoring events and visit events, achieving finer-grained relationships with clear medical significance.

**Temporal Feature Interaction.** Temporal feature interaction methods (Zheng et al., 2024; Feng et al., 2024) integrate temporal modelling into graph structures, allowing for the realistic representation of real-world systems by modelling changes over time. (1) The first type of research generates static graph sequences through temporal snapshots (Sankar et al., 2020; Wang et al., 2020; 2021c; Li et al., 2019; Jin et al., 2019; Qin et al., 2023), learning representations at each time point and integrating them sequentially using a temporal network. However, these methods only capture interactions within a single time point, neglecting feature interaction across multiple time steps. (2) Another approach continuously updates nodes and edges with timestamps, enabling smoother feature interaction and asynchronous time modelling (Trivedi et al., 2017; 2019; Han et al., 2020; Sun et al., 2022a). Some works focus on temporal models, while others (Wen & Fang, 2022; Ma et al., 2020b; Kumar et al., 2019; Zhang et al., 2024) focus on event intensity and edge order. Additionally, methods (Xu et al., 2020; Wang et al., 2021a;b; Li et al., 2023; Wu et al., 2024) use attention mechanisms and neighbour aggregation for asynchronous propagation. However, key dynamic features may fade quickly during edge adjustments, making it difficult to capture brief but crucial changes. This paper falls into the first category, leveraging the pathological relationships between monitoring and visit events across time points to achieve feature interaction.

## 3 METHOD

Our proposed method, CrossMed, as shown in Figure 3, consists of three distinct modules: (1) *Relationship Discovery*: Model pathological relationships between monitoring events and visit events. (2) *Graph Construction*: Establish a cross-level interaction temporal graph based on pathological relationships. (3) *Feature Interaction*: Perform feature interaction across different levels of events to generate patient representations.

### 3.1 MODULE 1: RELATIONSHIP DISCOVERY

To evaluate the influence of a monitoring event on a visit event, we define the specific monitoring event as the treatment variable $T$, the specific visit event as the outcome variable $Y$, and other related monitoring events as confounding variables $X$. We then apply a generalized linear model (GLM) with a logit link function, expressed as:

$$\log\left(\frac{\mu}{1-\mu}\right) = \beta_0 + \beta_T T + \beta_X X,  \tag{1}$$

where $\mu$ denotes the expected value of the outcome variable $Y$. In this model, $\beta_0$ represents the intercept, $\beta_T$ reflects the average effect of the treatment variable $T$ on the outcome variable $Y$, and

$\beta_X$ encompasses the coefficients for the confounding variable $X$. The parameters $\beta_0$, $\beta_T$, and $\beta_X$ are estimated using the maximum likelihood estimation (MLE) method. By fitting the estimated coefficient $\hat{\beta}_T$, we obtain the influence of a specific monitoring event on a specific visit event at a given time. Aggregating multiple $\hat{\beta}_T$ values yields $w_{m_{t,n}}^{m-D}$, $w_{m_{t,n}}^{m-P}$, and $w_{m_{t,n}}^{m-R}$, which represent the relationship effect weight between the monitoring event and disease, procedure, and drug at time $m_{t,n}$, respectively. For a detailed explanation, please refer to the Appendix B.2.

## 3.2 MODULE 2: GRAPH CONSTRUCTION

In this module, we construct a cross-level interaction temporal graph based on the dynamic pathological relationships between different levels of data, which is divided into two steps: node construction and edge construction.

**For node construction.** We generate four types of nodes from $\mathcal{N}^1$ to $\mathcal{N}^4$. The first type of node, $\mathcal{N}^1$, represents monitoring events, and $\mathbf{h}_{\mathcal{N}_{t,n}^1}$ denotes the representation of the monitoring event during the $n$-th monitoring session of the $t$-th visit. The second, third, and fourth types of nodes, $\mathcal{N}^2$, $\mathcal{N}^3$, and $\mathcal{N}^4$, all refer to visit events, representing diseases, procedures, and drugs, respectively. For $\mathbf{h}_{\mathcal{N}_{t,n}^2}$, it denotes the representation of the disease during the $n$-th monitoring session of the $t$-th visit. Since diseases are visit-level events, all $\mathbf{h}_{\mathcal{N}_{t,n}^2}$ within the same visit $V_t$ are initialized to be identical. Similarly, for $\mathbf{h}_{\mathcal{N}_{t,n}^3}$ and $\mathbf{h}_{\mathcal{N}_{t,n}^4}$, representing the procedure and drug at time of $m_{t,n}$, respectively, all $\mathbf{h}_{\mathcal{N}_{t,n}^3}$ and $\mathbf{h}_{\mathcal{N}_{t,n}^4}$ within the same visit $V_t$ are also initialized to be identical.

**For edge construction.** There are three types of edges in total in the cross-level interaction temporal graph in, as follows: (1) *Same-Time Same-Level Relationships* (blue dashed bi-directed edges): We model the direct link between visit events by constructing bi-directional edges between multiple visit events at the same time. Each edge is assigned a fixed weight of 1. (2) *Same-Time Cross-Level Relationships* (green dashed directed edges): We model the influence of monitoring events on visit events occurring at the same time point by constructing cross-level edges. The effects generated in the previous module ($w_{m_{t,n}}^{m-D}$, $w_{m_{t,n}}^{m-P}$, and $w_{m_{t,n}}^{m-R}$) are used as the corresponding edge weights. (3) *Cross-Time Relationships* (yellow solid directed edges): We model the response of visit events on monitoring events at the next time point by creating edges between consecutive time points. Furthermore, we construct edges between consecutive monitoring events to capture changes in health state over time, with each edge assigned a fixed weight of 1.

Notably, the graph structure mentioned above is used to aggregate multiple monitoring sessions into a visit. A similar graph is employed to aggregate multiple visits into a patient representation, as detailed in Appendix B.3.

## 3.3 MODULE 3: FEATURE INTERACTION

After building the cross-level interaction temporal graph, we perform feature interaction for multiple nodes based on the constructed edges.

**Dynamic edge weights.** We compute the dynamic edge weights $\eta$ according to the method proposed by GATv2 (Brody et al., 2022),

$$\eta_{ij}^{(r)} = \text{softmax}_j \left( \text{LeakyReLU}(\mathbf{a}^{(r)T}[\mathbf{W}^{(r)}\mathbf{h}_i \| \mathbf{W}^{(r)}\mathbf{h}_j]) \cdot w_{ij}^{(r)} \right), \tag{2}$$

where $\eta_{ij}^{(r)}$ represents the dynamic weight between nodes $i$ and $j$. The LeakyReLU function is a nonlinear activation function, while $\mathbf{a}^{(r)T}$ is a learnable attention vector specific to edge type $r$. $\mathbf{W}^{(r)}$ is the learnable weight matrix associated with edge type $r$, and $\mathbf{h}_i$ and $\mathbf{h}_j$ are the embedding representations of nodes $i$ and $j$, respectively. Finally, $w_{ij}^{(r)}$ is the weight assigned to the edge from node $j$ to node $i$ for edge type $r$ in the previous module.

**Feature interaction on same-time edges.** For edges within the same temporal sub-graph, we update node features as follows:

$$\mathbf{h}_i^{(l+1,t)} = (1-\alpha)\sigma \left( \sum_{r \in \mathcal{R}} \sum_{j \in \mathcal{N}_i^{(r)}} \eta_{ij}^{(r,l)} \mathbf{W}^{(r)} \mathbf{h}_j^{(l,t)} \right) + \alpha \mathbf{h}_i^{(l,t)}, \tag{3}$$

where $\mathbf{h}_j^{(l+1,t)}$ denotes the embedding of node $j$ at time step $t$ in layer $l+1$, $\mathcal{R}$ refers to the set of all edge types, $\alpha$ is the residual connection ratio, and $\sigma$ represents the ReLU activation function.

**Feature interaction on cross-time edges.** For edges across temporal sub-graphs, in addition to using the same graph network as for same-time edges, we also apply a temporal network method, as detailed below:

$$\mathbf{h}_i^{(l+1,t+1)} = (1 - \mathbf{z}_i^{(t+1)}) \odot \mathbf{h}_i^{(l+1,t)} + \mathbf{z}_i^{(t+1)} \odot \tanh\left(\mathbf{U}_h(\mathbf{r}_i^{(t+1)} \odot \mathbf{h}_i^{(l+1,t)})\right), \quad (4)$$

where $\mathbf{z}_i^{(t+1)}$ is the update gate controlling how much of the previous hidden state is kept, $\mathbf{U}_h$ is a learnable weight matrix that transforms the reset-modified hidden state, and $\mathbf{r}_i^{(t+1)}$ is the reset gate controlling how much of the previous hidden state contributes to the new candidate state. Finally, we derive the representations for the last time point: $\mathbf{h}_{\mathcal{N}_{t,N}^1}, \mathbf{h}_{\mathcal{N}_{t,N}^2}, \mathbf{h}_{\mathcal{N}_{t,N}^3}, \mathbf{h}_{\mathcal{N}_{t,N}^4}$.

Each cross-level temporal graph captures the representation of the last time point within its respective sequence. In the monitoring-to-visit stage, we derive the representations of the last monitoring session and concatenate these representations to obtain the visit representation, $\mathbf{h}_{V_t}$. Similarly, in the visit-to-patient stage, we derive the representations of the last visit and concatenate them into the patient representation, $\mathbf{h}_H$, which serves as the final representation for downstream tasks.

### 3.4 PREDICTION, TRAINING AND INFERENCE

**Prediction.** Based on the patient representation $h_H$, we produce outputs for various tasks using different predictors tailored to each task,

$$\mathbf{o}_{dp} = \text{sigmoid}(\text{fc}_{dp}(\mathbf{h}_H)), \quad \mathbf{o}_{dr} = \text{sigmoid}(\text{fc}_{dr}(\mathbf{h}_H)), \quad (5)$$

where $\mathbf{o}_{dp}/\mathbf{o}_{dr}$ represents the probability for each disease and drug, and $\text{fc}_{dp}/\text{fc}_{dr}$ is the independent predictor for the disease prediction and drug recommendation. Finally, we output the diseases/drugs with probabilities greater than 0.5.

**Training & Inference.** During the training phase, we optimize all the learnable parameters and use the same loss function for both tasks. The model follows the same pipeline during inference as it does in training. We denote the predicted label as $y_i$, and probability as $o_i$. The loss function, binary cross-entropy (BCE) loss, is used to optimize the model across both tasks, which is expressed as:

$$\mathcal{L}_{bce} = -\frac{1}{|X|} \sum_{i=1}^{|Y|} [y_i \log(o_i) + (1 - y_i) \log(1 - o_i)]. \quad (6)$$

## 4 EXPERIMENTS

### 4.1 EXPERIMENTAL SETTING

**Dataset.** This paper utilizes two widely used datasets, MIMIC-III (Johnson et al., 2016) and MIMIC-IV (Johnson et al., 2023). For a detailed description of the data pre-process, please refer to the Appendix C.1.

**Baselines.** To validate our model, we selected the following state-of-the-art benchmark models for comparison. For disease prediction, we selected RETAIN (Choi et al., 2016), Transformer (Vaswani, 2017), KAME (Ma et al., 2018), StageNet (Gao et al., 2020), REFINE (Bhoi et al., 2024), and TRANS (Chen et al., 2024). For drug recommendation, we selected RETAIN, Transformer, Grasp (Zhang et al., 2021), GAMENet (Shang et al., 2019b), SafeDrug (Yang et al., 2021b), Micron (Yang et al., 2021a), MoleRec (Yang et al., 2023b), REFINE, TRANS, and CausalMed (Li et al., 2024). Notably, we also introduced an MLP baseline that incorporates both visit and monitoring information for both tasks. For a detailed description of the baselines, please refer to the Appendix C.2.

**Evaluation Metrics.** To comprehensively evaluate our model, we used both the medical system and recommender system evaluation methods. For the medical system, we use four main general metrics (according to Jiang et al. (2023)) to evaluate the performance of our method: the F1-score, Jaccard, PR-AUC, and ROC-AUC. For the recommender system, we use the visit-level precision@k and event-level accuracy@k (according to Chen et al. (2024)) to evaluate our methods. For a detailed description of the evaluation metrics, please refer to the Appendix C.3.

Table 1: The average performance (%) and standard deviation (in parentheses) of each model for MIMIC-III and MIMIC-IV on both tasks, evaluated using medical system metrics. The top model is in **bold**, and the second-best is underlined, and models marked with an asterisk (*) indicate significance testing against the current state-of-the-art.

| | Disease Prediction | | | | | | | |
|---|---|---|---|---|---|---|---|---|
| **Model** | MIMIC-III | | | | MIMIC-IV | | | |
| | F1-score | Jaccard | PR-AUC | ROC-AUC | F1-score | Jaccard | PR-AUC | ROC-AUC |
| RETAIN | 36.22 (1.4) | 22.11 (1.0) | 48.97 (1.2) | 92.35 (0.3) | 37.43 (1.1) | 23.02 (0.8) | 47.48 (1.0) | 91.69 (0.2) |
| Transformer | 35.26 (1.1) | 21.40 (0.8) | 45.85 (1.0) | 91.58 (0.4) | 35.97 (4.1) | 24.94 (2.6) | 44.38 (3.8) | 90.20 (0.9) |
| KAME | 34.21 (1.3) | 20.64 (0.9) | 42.76 (1.3) | 90.12 (0.3) | 37.41 (1.2) | 23.01 (0.9) | 45.51 (1.3) | 90.88 (0.4) |
| StageNet | 41.58 (1.0) | 26.05 (0.8) | 44.36 (1.1) | 90.65 (0.4) | 44.49 (0.9) | 28.61 (0.8) | 47.60 (1.3) | 91.34 (0.3) |
| Trans | 38.55 (1.9) | 23.88 (1.5) | 49.62 (2.6) | 92.45 (0.3) | 38.61 (1.5) | 23.93 (1.2) | 51.68 (1.2) | 92.66 (0.2) |
| MLP | 37.21 (1.0) | 23.17 (0.8) | 42.13 (1.8) | 89.93 (0.4) | 40.25 (0.8) | 24.78 (0.7) | 45.43 (1.6) | 90.74 (0.4) |
| REFINE | 38.84 (1.9) | 24.10 (1.4) | 49.53 (1.2) | 92.09 (0.4) | 41.14 (1.1) | 25.90 (0.4) | 51.80 (1.3) | 92.66 (0.2) |
| **Ours** | **43.31 (1.4)\*** | **27.80 (1.1)\*** | **51.37 (1.4)\*** | **93.43 (0.4)\*** | **46.82 (1.7)\*** | **29.78 (0.9)\*** | **52.16 (1.3)** | **93.10 (0.6)\*** |
| | Drug Recommendation | | | | | | | |
| **Model** | MIMIC-III | | | | MIMIC-IV | | | |
| | F1-score | Jaccard | PR-AUC | ROC-AUC | F1-score | Jaccard | PR-AUC | ROC-AUC |
| RETAIN | 60.26 (3.3) | 43.12 (3.4) | 71.61 (3.8) | 90.87 (1.2) | 62.18 (1.8) | 45.12 (1.9) | 73.25 (2.0) | 91.81 (0.5) |
| Transformer | 59.75 (1.3) | 42.52 (1.5) | 73.95 (3.0) | 92.08 (0.9) | 58.11 (3.5) | 40.96 (3.5) | 70.22 (3.7) | 90.86 (1.2) |
| Grasp | 62.02 (2.3) | 47.49 (2.0) | 76.16 (2.2) | 92.89 (1.5) | 63.01 (2.4) | 47.53 (2.3) | 75.98 (2.7) | 91.75 (1.5) |
| GAMENet | 63.43 (2.4) | 48.94 (1.9) | 77.63 (2.1) | 93.40 (1.6) | 63.78 (2.6) | 49.33 (2.2) | 78.31 (2.5) | 93.65 (1.6) |
| SafeDrug | 59.60 (2.2) | 45.04 (2.0) | 75.46 (2.4) | 92.34 (1.5) | 59.59 (3.6) | 44.90 (2.2) | 75.34 (2.7) | 92.20 (1.6) |
| Micron | 61.71 (2.3) | 46.98 (2.0) | 75.05 (2.3) | 92.61 (1.7) | 62.79 (2.7) | 48.33 (2.4) | 77.12 (2.6) | 93.18 (1.8) |
| MoleRec | 64.44 (2.7) | 49.62 (2.8) | 76.77 (2.4) | 92.63 (1.7) | 64.85 (1.8) | 50.18 (2.6) | 76.88 (2.7) | 92.11 (1.8) |
| Trans | 63.49 (3.0) | 46.44 (3.2) | 75.73 (3.0) | 91.95 (0.9) | 64.13 (2.4) | 47.13 (2.6) | 76.14 (3.0) | 92.37 (0.9) |
| CausalMed | 66.14 (2.5) | 51.29 (2.0) | 79.00 (1.8) | 93.11 (0.6) | 66.27 (2.5) | 50.27 (2.3) | 78.56 (2.4) | 93.09 (1.7) |
| MLP | 63.64 (2.1) | 46.67 (2.3) | 69.72 (2.3) | 89.38 (0.9) | 62.75 (2.2) | 47.87 (2.4) | 67.88 (4.5) | 90.37 (1.2) |
| REFINE | 66.73 (2.6) | 50.07 (2.9) | 78.25 (2.6) | 92.54 (0.7) | 66.05 (1.0) | 49.66 (1.1) | 78.29 (1.5) | 92.11 (0.3) |
| **Ours** | **69.58 (2.3)** | **53.35 (2.7)** | **79.02 (1.8)** | **93.61 (0.6)** | **68.74 (2.5)\*** | **52.37 (2.8)** | **79.34 (2.5)** | **94.12 (1.7)\*** |

Table 2: The average performance (%) and standard deviation (in parentheses) of each model for MIMIC-III and MIMIC-IV on both tasks, evaluated using recommender system metrics. The top model is in **bold**, and the second-best is underlined, and models marked with an asterisk (*) indicate significance testing against the current state-of-the-art.

| | Disease Prediction | | | | | | | | | | | |
|---|---|---|---|---|---|---|---|---|---|---|---|---|
| **Model** | MIMIC-III | | | | | | MIMIC-IV | | | | | |
| | Event-Level Accuracy@k | | | Visit-Level Precision@k | | | Event-Level Accuracy@k | | | Visit-Level Precision@k | | |
| | 10 | 20 | 30 | 10 | 20 | 30 | 10 | 20 | 30 | 10 | 20 | 30 |
| RETAIN | 22.79 (2.3) | 25.58 (1.7) | 26.23 (3.0) | 30.80 (2.5) | 25.70 (1.8) | 26.61 (2.4) | 21.10 (2.0) | 23.77 (1.5) | 24.22 (3.1) | 37.09 (2.2) | 26.20 (1.7) | 27.67 (2.8) |
| Transformer | 21.66 (1.3) | 24.19 (2.1) | 26.07 (2.7) | 30.26 (1.8) | 25.13 (2.4) | 26.47 (2.0) | 20.86 (1.7) | 24.26 (2.3) | 26.47 (2.8) | 32.36 (1.9) | 24.89 (2.5) | 25.79 (1.8) |
| KAME | 24.90 (2.0) | 25.32 (1.6) | 27.91 (3.0) | 32.50 (2.2) | 26.97 (2.7) | 27.28 (1.8) | 25.00 (2.1) | 28.82 (1.9) | 29.56 (2.4) | 34.54 (1.5) | 27.66 (2.2) | 28.71 (2.9) |
| StageNet | 26.50 (1.8) | 27.51 (2.3) | 28.89 (2.7) | 33.80 (1.9) | 29.91 (2.4) | 30.72 (2.1) | 27.41 (1.6) | 29.19 (2.2) | 31.37 (2.9) | 38.63 (2.0) | 29.57 (1.9) | 31.03 (2.7) |
| Trans | 25.31 (1.2) | 28.02 (1.3) | 29.77 (1.4) | 35.10 (1.4) | 29.16 (1.2) | 30.40 (1.2) | 27.33 (1.2) | 32.23 (1.2) | 34.66 (1.2) | 42.95 (1.2) | 34.20 (1.2) | 34.74 (1.2) |
| MLP | 22.73 (1.0) | 23.41 (1.3) | 26.58 (1.4) | 33.55 (1.6) | 26.09 (1.6) | 27.41 (1.7) | 26.31 (0.7) | 29.69 (0.9) | 30.29 (0.9) | 38.52 (1.8) | 31.49 (1.2) | 32.29 (1.2) |
| REFINE | 23.61 (1.4) | 25.90 (1.1) | 27.75 (1.1) | 33.14 (1.4) | 27.29 (1.4) | 28.62 (1.6) | 27.15 (1.0) | 31.72 (0.9) | 33.88 (1.2) | 42.29 (1.9) | 32.80 (0.8) | 33.33 (1.3) |
| **Ours** | **29.36 (2.5)** | **32.84 (2.3)\*** | **34.66 (2.8)\*** | **40.74 (2.1)\*** | **33.66 (1.9)\*** | **34.82 (2.2)\*** | **27.96 (2.0)** | **33.33 (2.1)** | **35.61 (2.4)** | **43.71 (1.9)** | **34.64 (2.3)** | **35.05 (2.7)** |
| | Drug Recommendation | | | | | | | | | | | |
| **Model** | MIMIC-III | | | | | | MIMIC-IV | | | | | |
| | Event-Level Accuracy@k | | | Visit-Level Precision@k | | | Event-Level Accuracy@k | | | Visit-Level Precision@k | | |
| | 30 | 40 | 50 | 30 | 40 | 50 | 30 | 40 | 50 | 30 | 40 | 50 |
| RETAIN | 46.75 (2.3) | 51.17 (1.7) | 56.87 (3.1) | 64.38 (2.0) | 64.14 (2.2) | 64.20 (2.5) | 44.71 (1.8) | 49.45 (2.1) | 52.41 (3.0) | 61.34 (2.0) | 64.28 (2.3) | 63.36 (2.8) |
| Transformer | 47.43 (1.4) | 52.91 (1.2) | 56.78 (0.8) | 64.61 (1.1) | 64.19 (0.7) | 64.92 (0.7) | 45.91 (1.3) | 50.43 (1.1) | 53.70 (1.1) | 63.66 (1.6) | 62.06 (1.3) | 62.53 (1.0) |
| Grasp | 46.71 (1.9) | 53.28 (2.4) | 57.66 (2.5) | 65.48 (2.0) | 63.54 (2.3) | 64.76 (2.4) | 46.16 (2.1) | 52.03 (2.3) | 55.04 (2.4) | 64.07 (2.1) | 66.09 (2.5) | 65.05 (2.9) |
| GAMENet | 47.72 (2.2) | 54.32 (2.5) | 58.94 (2.1) | 66.27 (2.3) | 64.92 (2.1) | 65.33 (2.6) | 47.16 (2.0) | 54.74 (2.4) | 56.89 (2.9) | 65.65 (2.2) | 66.43 (2.1) | 66.40 (2.7) |
| SafeDrug | 46.17 (2.2) | 52.62 (2.5) | 59.83 (2.9) | 63.39 (2.3) | 62.54 (2.0) | 63.81 (2.7) | 46.26 (1.9) | 52.65 (2.2) | 54.75 (2.5) | 62.32 (2.6) | 63.19 (2.4) | 63.55 (2.3) |
| Micron | 47.67 (2.1) | 51.87 (2.0) | 54.75 (2.3) | 65.55 (1.8) | 64.69 (2.4) | 64.70 (2.1) | 45.58 (1.8) | 52.11 (2.4) | 55.31 (2.6) | 64.54 (2.5) | 63.24 (2.8) | 64.25 (2.6) |
| MoleRec | 49.94 (2.7) | 58.89 (2.4) | 61.28 (2.2) | 65.78 (2.5) | 65.53 (2.3) | 65.86 (2.9) | 47.81 (2.4) | 52.13 (2.5) | 57.87 (2.6) | 64.22 (2.1) | 64.50 (2.8) | 65.01 (2.3) |
| Trans | 47.15 (2.9) | 52.33 (2.9) | 56.35 (3.4) | 64.98 (1.5) | 64.12 (1.8) | 65.00 (2.2) | 47.05 (2.1) | 51.61 (2.1) | 55.16 (2.4) | 65.66 (2.5) | 63.91 (1.9) | 64.37 (2.0) |
| CausalMed | 51.32 (2.5) | 56.45 (2.7) | 61.17 (2.1) | 66.02 (2.6) | 64.13 (2.4) | 66.24 (2.8) | 46.33 (2.3) | 53.41 (2.2) | 59.14 (2.8) | 65.04 (2.5) | 67.21 (2.4) | 66.18 (2.9) |
| MLP | 44.75 (1.5) | 51.75 (1.7) | 56.76 (1.9) | 64.45 (2.2) | 63.39 (1.7) | 63.26 (1.6) | 45.57 (1.8) | 51.12 (1.5) | 56.90 (1.4) | 64.09 (2.1) | 63.28 (1.6) | 64.57 (1.4) |
| REFINE | 47.95 (1.6) | 53.64 (1.2) | 58.04 (0.9) | 64.11 (1.3) | 63.42 (0.7) | 65.85 (0.4) | 48.39 (1.4) | 53.93 (1.3) | 58.02 (0.8) | 66.32 (0.9) | 65.07 (0.6) | 65.68 (0.8) |
| **Ours** | **55.18 (2.3)\*** | **59.72 (2.8)\*** | **63.99 (2.4)\*** | **67.19 (2.6)** | **66.32 (2.9)** | **67.13 (2.5)** | **47.96 (2.1)** | **55.58 (2.4)\*** | **61.65 (2.7)** | **67.89 (2.6)\*** | **68.16 (2.5)** | **69.43 (2.8)** |

## 4.2 Results and Analysis

In this section, we compare CrossMed to the baseline disease prediction and drug recommendation tasks and conduct several complementary experiments (some additional experiments are discussed in Appendix D) designed to answer the following research question (RQ).

*RQ1*: Does CrossMed provide more accurate clinical prediction than SOTA models for both tasks?

*RQ2*: Do the components we proposed improve the performance for both tasks?

*RQ3*: How does CrossMed perform with limited sequence length of visit and monitoring?

*RQ4*: How do different feature interaction learning methods impact performance, and why?

Table 3: Ablation experiments results (%) and standard deviation (in parentheses) of modified model for MIMIC-III on both tasks, evaluated using medical system metrics. The top model is in **bold**.

| Model | Disease prediction | | | | Drug recommendation | | | |
|---|---|---|---|---|---|---|---|---|
| | F1-score | Jaccard | PR-AUC | ROC-AUC | F1-score | Jaccard | PR-AUC | ROC-AUC |
| CrossMed w/o $R_{m2v}$ | 42.50 (1.2) | 26.93 (1.0) | 50.72 (1.3) | 92.80 (0.5) | 67.98 (2.1) | 52.36 (2.6) | 77.23 (1.8) | 92.42 (0.8) |
| CrossMed w/o $R_{v2m}$ | 43.05 (1.3) | 27.23 (1.0) | 50.04 (1.3) | 91.72 (0.5) | 68.24 (2.3) | 51.70 (2.3) | 77.05 (1.7) | 91.49 (0.9) |
| CrossMed w/o $RM$ | 43.28 (1.4) | 27.32 (1.1) | 50.05 (1.4) | 92.93 (0.5) | 68.62 (1.9) | 52.97 (2.5) | 78.87 (2.0) | 92.97 (0.8) |
| CrossMed w/o $TR$ | 41.17 (1.2) | 26.15 (0.9) | 48.40 (1.2) | 92.25 (0.6) | 65.50 (2.1) | 50.40 (2.6) | 75.24 (1.4) | 90.63 (0.9) |
| **CrossMed** | **43.31** (1.4) | **27.80** (1.1) | **51.37** (1.4) | **93.43** (0.4) | **69.58** (2.3) | **53.35** (2.7) | **79.02** (1.8) | **93.61** (0.6) |

**Performance Comparison (RQ1).** Tables 1 and 2 demonstrate the performance of the CrossMed model proposed in this paper with other baseline models under two datasets, two tasks, and two sets of evaluation metrics systems. Models like Trans and CausalMed, which emphasize relationships between visit events and perform feature interactions, outperform models like Micron and StageNet which focus mainly on the temporal dependencies within visit sequences.

Additionally, REFINE utilizes monitoring sequences, further improving performance compared to methods that only use visit sequences. These advantages arise because performing feature interaction between multiple medical events can significantly enhance the accuracy of event representations. Moreover, monitoring sequences are finer-grained than visit sequences, and modelling the temporal relationships within monitoring allows for capturing clearer patient health trends. Our method models the pathological relationships between monitoring events and visit events, capturing finer-grained health states and enabling feature interactions between the two sequences. Compared to the baseline models, our CrossMed achieves superior performance across both datasets, tasks, and evaluation systems.

**Ablation Study (RQ2).** We conducted an ablation study, as shown in Table 3, to evaluate the effectiveness of each CrossMed component by removing four key elements: the relationship from monitoring to visit events ($R_{m2v}$), the relationship from visit to monitoring events ($R_{v2m}$), the relationship discovery module ($RM$), and the temporal recurrent component in feature interaction ($TR$). Removing $R_{m2v}$ and $R_{v2m}$ significantly reduces model accuracy, underscoring the importance of capturing interactions between different-level events. Excluding $RM$ also results in suboptimal performance, indicating the necessity of modelling the granular impact of monitoring events on visit events. Omitting $TR$ causes a performance decline, demonstrating the critical role of temporal propagation in tracking patient health trends. Overall, the core methods proposed by CrossMed are essential for improving model effectiveness.

**Robustness Study (RQ3).** We evaluated CrossMed's performance with limited visit and monitoring sequence lengths through a robustness study on drug recommendation tasks using the MIMIC-III dataset, assessed by F1-score. We create scenarios by limiting visits per patient and monitoring per visit.

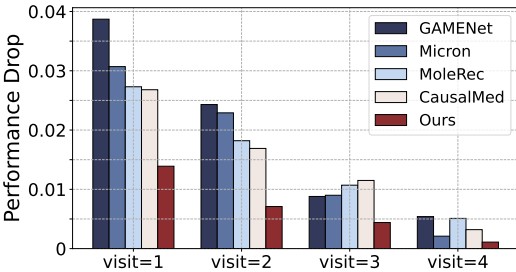

Figure 4: The performance decrease of different models with different limited lengths of visit sequence compared to the optimal performance, where higher bars represent more decrease.

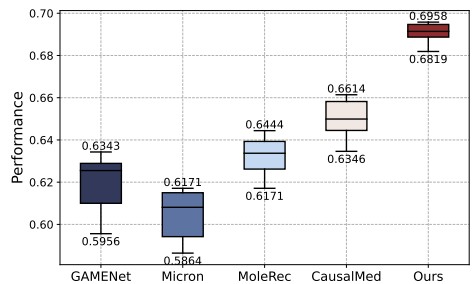

Figure 5: Robustness of various models with a limited length of visit sequence, where larger boxes represent more significant impacts.

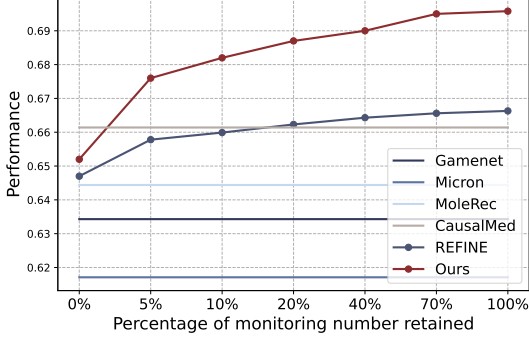

Figure 6: The performance of our method in different scenarios with a limited length of monitoring sequence. Some models do not use the monitoring sequence, so there was no change.

*For the length of visit sequence:* Figure 4 shows model performance with varying visit numbers. While GAMENet and Micron struggle with low visit numbers, MoleRec, CausalMed, and CrossMed remain stable by focusing on feature interactions beyond visit sequences. Figure 5 highlights CrossMed's smaller variability and superior performance due to its integration of monitoring and visit data, capturing finer-grained health trends.

*For the length of monitoring sequence:* Figure 6 shows model performance as monitoring sequence length decreases. Even with 0% sequence length (*i.e.*, retaining monitoring event nodes without embedded information), our model slightly outperforms others. Compared to RE-FINE, our cross-level interaction method shows greater improvement as sequence length increases, excelling in both ICU settings with long monitoring sequences and routine predictions.

**Feature Interaction Method Study (RQ4).** To validate the effectiveness of our cross-level feature interaction method, we conducted comparative experiments on the MIMIC-III dataset, evaluating five interaction methods: (1) visit sequences only, (2) parallel modelling of visit and monitoring sequences, (3) modelling the influence of monitoring on visits, (4) modelling the influence of visits on monitoring, and (5) cross-level feature interactions. As shown in Figure 7, the parallel interaction method performs similarly to the visit-only method, both yielding sub-optimal results compared to strong baselines (StageNet and CausalMed). In contrast, cross-level interaction methods significantly improve performance by effectively capturing pathological relationships. To further

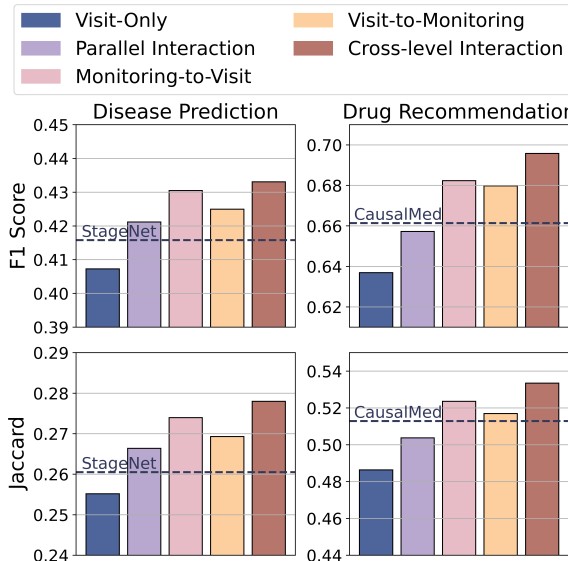

Figure 7: The impact of different data interaction methods on performance for both tasks, where the dashed line represents the performance of the sub-optimal model.

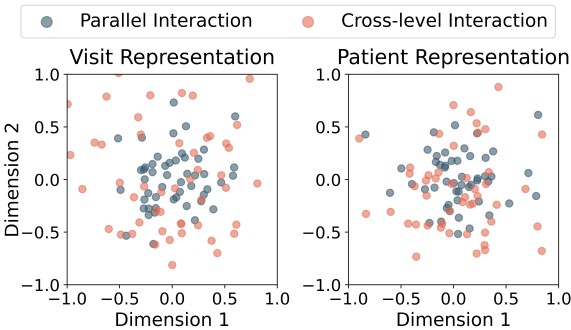

Figure 8: The t-SNE visualization shows the distances between representations generated by different feature interaction methods. The larger the distance, the higher the differentiation of the representation.

explore the effectiveness of cross-level interactions, t-SNE visualizations, as shown in Figure 8, show that cross-level interactions produce more distinct and well-separated representations, confirming that CrossMed captures finer-grained patient health trends, enhancing clinical prediction by improving the differentiation of visit and patient representations.

## 5    CONCLUSION

This paper introduces CrossMed, a structured EHRs modelling method for disease prediction and drug recommendation. By modelling dynamic pathological relationships and using a novel cross-level feature interaction approach, CrossMed effectively captures patient health trends during treatment. Experiments on two public medical datasets show it outperforms all baselines. Although CrossMed improves prediction accuracy, it currently captures relationships between medical events based on simple correlations. However, these pathological relationships are often much more complex in reality. Future work aims to better model these relationships using more advanced methods to enhance this framework.

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

## A PRELIMINARIES DETAILS

### A.1 DATA FORMAT DETAILS

For each clinical visit $V_t$, we have $V_t = \{S_t, D_t, P_t, R_t, M_t\}$, where $S_t$, $D_t$, $P_t$, $R_t$, $M_t$ represent the covariates, diseases, procedures, drugs, and monitoring information of the patient, respectively. The diseases, procedures, and drugs have already been introduced in the main text; here, we will elaborate on the covariates $S_t$ and monitoring $M_t$.

**Covariates information** Covariates information $S_t$ includes age and weight of the patient at the time of visit $V_t$ in this paper, represented as $S_t = \{\text{age}_t, \text{wgt}_t\}$, where $\text{age}_t \in [0, 1]$ and $\text{wgt}_t \in [0, 1]$ are both normalized continuous values, representing the patient's age and weight, respectively.

**Monitoring information** For a single monitoring session, we have $m_{t,n} = \{\text{lab}_{t,n}, \text{inj}_{t,n}\}$, where $\text{lab}_{t,n}$ and $\text{inj}_{t,n}$ are both monitoring events, represent the laboratory results and injection dosages of a patient during the $n$-th monitoring session of the $t$-th visit, respectively. For the laboratory results, we have $\text{lab}_{t,n} = \{\text{lab}^1_{t,n}, \text{lab}^2_{t,n}\}$, where $\text{lab}^1_{t,n}$ refers to multiple lab test items performed in the same monitoring session. It is represented in multi-hot encoding form, i.e., $\text{lab}^1_{t,n} \in \{0, 1\}^{|LAB|}$, with $|LAB|$ denoting the total number of categories of lab test items. A value of 1 indicates that the test was performed, while a value of 0 indicates that it was not. $\text{lab}^2_{t,n} \in \{0, 1\}^{|LAB|}$ represents the results of the performed lab tests in the same session, also encoded in multi-hot form, where 1 denotes an abnormal result and 0 denotes a normal result. For the injection dosage, we have $\text{inj}_{t,n} = \{\text{inj}^1_{t,n}, \text{inj}^2_{t,n}\}$, where $\text{inj}^1_{t,n}$ refers the multiple injection items in the monitoring session, encoded similarly to $\text{lab}^1_{t,n}$, representing $\text{inj}^1_{t,n} \in \{0, 1\}^{|INJ|}$, with $|INJ|$ denoting the total number of categories of injection items. Meanwhile, $\text{inj}^2_{t,n} \in [0, 1]^{|INJ|}$ indicates the dosage of the injections in the same monitoring, expressed as a normalized vector.

### A.2 NOTATIONS

Important mathematical notes can be found in Table 4.

Table 4: Mathematical Notations

| Notations | Descriptions |
|---|---|
| $H, V_t$ | patient, the $t$-th visit |
| $C_t, c$ | disease set in $V_t$, a disease |
| $P_t, p$ | procedure set in $V_t$, a procedure |
| $D_t, d$ | drug set in $V_t$, a drug |
| $S_t$ | covariates in $V_t$ |
| $\text{age}_t, \text{wgt}_t$ | age, weight in $V_t$ |
| $M_t, m_{t,n}$ | monitoring sequence in $V_t$, the $n$-th monitoring in $V_t$ |
| $\text{lab}_{t,n}, \text{inj}_{t,n}$ | laboratory results, injection dosage in $m_{t,n}$ |
| $E, \mathbf{h}$ | embedding table and representation |
| $T, Y, X$ | treatment, outcome, and confounding variable |
| $\mu, \beta$ | value of outcome variable and linear coefficient |
| $w$ | pathological relationship weight |
| $\mathcal{E}, \mathcal{N}$ | edge and node of graph |
| $r$ | edge type |
| $\mathbf{a}, \mathbf{W}$ | learnable attention vector, learnable weight matrix |
| $e, \eta$ | attention score and attention coefficient |
| $\alpha$ | ratio of residual connections |
| $U, \mathbf{z}, \mathbf{r}$ | weight matrix of hidden states, update gate, reset gate |
| $t, l$ | time step, graph layer |

Figure 9: A specific example of the Relationship Discovery module is uncovering the influence of monitoring events (lab tests) on visit events (diseases, procedures, and drugs).

## B METHODOLOGY DETAILS

### B.1 REPRESENTATION INITIALIZATION

**For covariates.** We use two feed forward networks: $\mathrm{fc}_{age}(\cdot) : \mathbb{R}^1 \to \mathbb{R}^{\dim}$ and $\mathrm{fc}_{wgt}(\cdot) : \mathbb{R}^1 \to \mathbb{R}^{\dim}$ to characterize continuous values of age $age$ and weight $wgt$:

$$\mathbf{h}_{\text{age}} = \mathrm{fc}_{age}(age), \quad \mathbf{h}_{\text{wgt}} = \mathrm{fc}_{wgt}(wgt). \tag{7}$$

**For visit events.** We define three learnable embedding tables $E_d \in \mathbb{R}^{|D|\times\dim}$, $E_p \in \mathbb{R}^{|P|\times\dim}$ and $E_r \in \mathbb{R}^{|R|\times\dim}$, corresponding to disease, procedure, and drug, where dim is the embedding dimension. As shared representations in global data, $\mathbf{h}_{d_i}$, $\mathbf{h}_{p_j}$, and $\mathbf{h}_{r_k}$ are generated by mapping the disease $d_i$, the procedure $p_j$, and the drug $r_k$ into the embedding space:

$$\mathbf{h}_{d_i} = d_i E_d, \quad \mathbf{h}_{p_j} = p_j E_p, \quad \mathbf{h}_{r_k} = r_k E_r. \tag{8}$$

Then we perform additive aggregation on all diseases, procedures, and drugs in visit $V_t$ to obtain $\mathbf{h}_{D_t}$, $\mathbf{h}_{P_t}$, and $\mathbf{h}_{R_t}$, respectively.

**For monitoring events.** We similarly define two sets of embedding tables: $E_{\text{lab}}^1 \in \mathbb{R}^{|\text{LAB}|\times\dim}$ and $E_{\text{lab}}^2 \in \mathbb{R}^{|\text{LAB}|\times\dim}$, as well as $E_{\text{inj}}^1 \in \mathbb{R}^{|\text{INJ}|\times\dim}$ and $E_{\text{inj}}^2 \in \mathbb{R}^{|\text{INJ}|\times\dim}$. Where $E_{\text{lab}}^1$ corresponds to the lab test item, $E_{\text{lab}}^2$ corresponds to the lab result, $E_{\text{inj}}^1$ corresponds to the injection item, and $E_{\text{inj}}^2$ corresponds to the injected dosage. The lab test information and injection information are generated by combining the two sets of information:

$$\mathbf{h}_{\text{lab}_i} = \text{lab}_i^1 E_{\text{lab}}^1 \cdot \text{lab}_i^2 E_{\text{lab}}^2, \quad \mathbf{h}_{\text{inj}_j} = \text{inj}_j^1 E_{\text{inj}}^1 \cdot \text{inj}_j^2 E_{\text{inj}}^2, \tag{9}$$

where $\text{lab}_i^1$ represents the laboratory test that the patient underwent, while $\text{lab}_i^2$ records the specific result of that test. Similarly, in the monitoring records for injection j, $\text{inj}_j^1$ indicates the injection that the patient received, and $\text{inj}_j^2$ denotes the dosage of that injection. Then we perform additive aggregation on all lab tests and injection dosage in $m_{t,n}$ to obtain $\mathbf{h}_{\text{LAB}_{t,n}}$ and $\mathbf{h}_{\text{INJ}_{t,n}}$, respectively.

### B.2 DETAILS IN MODULE1: RELATIONSHIP DISCOVERY

We illustrate the specific process of the relationship discovery module using an example, shown in Figure 9, that captures the influence of a particular monitoring event (lab test) on three types of visit events (disease, procedure, drug).

**Input**: Data from all patients in the EHR.

**Effect Estimation**: As described in the main text, a linear model is used to capture the general associations between monitoring events and visit events.

**Effect Aggregation**: The edge weights of each pair of monitoring and visit events are aggregated using average pooling, forming an influence effect of the monitoring event set on the visit event set.

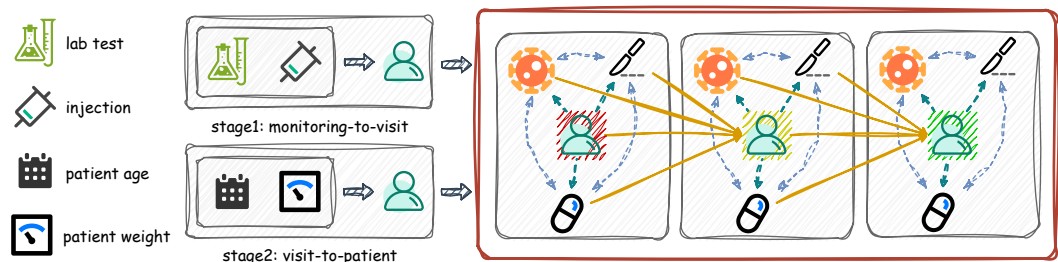

Figure 10: A similar graph construction method is used in both the monitoring-to-visit aggregation and the visit-to-patient aggregation processes.

**Output**: Pathological relationships between the lab test set and various visit event sets.

This paper repeatedly utilizes the Relationship Discovery module to generate the influence relationships of monitoring events (lab tests, injection dosages) visit events (diseases, procedures, drugs) and covariates (age, weight) on visit events.

### B.3 Details in Module2: Construction of Cross-level Temporal Graph

This module can be reused in both the aggregating monitoring-to-visit and aggregating visit-to-patient stages, with the main difference lying in node construction.

**For monitoring-to-visit stage**: $\mathcal{N}_{t,n}^1$ represents the monitoring visit, while $\mathcal{N}_{t,n}^2$, $\mathcal{N}_{t,n}^3$, and $\mathcal{N}_{t,n}^4$ represent diseases, procedures, and medications, respectively. Since they all belong to the same $V_t$, the initial representations of nodes representing visit events are the same across different monitoring sessions.

**For visit-to-patient stage**: $\mathcal{N}_t^1$, $\mathcal{N}_t^2$, $\mathcal{N}_t^3$, and $\mathcal{N}_t^4$ represent covariates, diseases, procedures, and medications in $V_t$, respectively, with the initial representations of visit event nodes changing over time.

### B.4 Details in Inference

During the inference phase, the model works within the same pipeline as training. We use the parameters trained on the training set to perform inference on the validation set. The model that achieves the lowest loss on the validation set is considered to have the best performance, and its parameters are selected as the optimal ones.

## C Experimental Setup Details

### C.1 Dataset and Data Pre-process

**Dataset.** The dataset used for both tasks is the same. As shown in Table 5, this paper utilizes the MIMIC-III[2] (Johnson et al., 2016) and MIMIC-IV[3] (Johnson et al., 2023) datasets, which are widely used in clinical research and analysis. The codes involved in this paper and datasets include the ICD-9[4], ICD-10[5], CCS[6], NDC[7], and ATC[8]. In the MIMIC-III dataset, diseases are encoded using ICD-9-CM codes, while in the MIMIC-IV dataset, both ICD-9-CM and ICD-10-CM codes

---

[2]https://physionet.org/content/mimiciii/1.4/

[3]https://physionet.org/content/mimiciv/3.0/

[4]https://www.cms.gov/medicare/coding-billing/icd-10-codes

[5]https://www.cms.gov/medicare/coding-billing/icd-10-codes/
icd-9-cm-diagnosis-procedure-codes-abbreviated-and-full-code-titles

[6]https://www.nlm.nih.gov/research/umls/sourcereleasedocs/current/CCS

[7]https://www.fda.gov/drugs/drug-approvals-and-databases/
national-drug-code-directory

[8]https://www.who.int/tools/atc-ddd-toolkit/atc-classification

are used. In this paper, these codes are unified and mapped to CCS-CM codes. For procedures, the MIMIC-III dataset uses ICD-9-PROC codes, while the MIMIC-IV dataset uses both ICD-9-PROC and ICD-10-PROC codes, which are unified and mapped to CCS-PROC codes in this study. Drugs in both the MIMIC-III and MIMIC-IV datasets are encoded using NDC codes and are mapped to ATC-3 codes in this paper.

Table 5: Statistics of the datasets.

| Items | MIMIC-III | MIMIC-IV |
|---|---|---|
| #num. of patients | 15,407 | 19,721 |
| #num. of visits | 18,557 | 24,777 |
| #num. of diseases | 272 | 276 |
| #num. of procedures | 204 | 213 |
| #num. of drugs | 196 | 200 |
| #num. of lab item | 669 | 807 |
| #num. of inj. item | 279 | 309 |
| #avg. of visits/ patient | 1.2950 | 1.2564 |
| #avg. of dis./ visit | 12.7193 | 14.7674 |
| #avg. of proc./ visit | 3.3714 | 3.2920 |
| #avg. of drug/ visit | 34.2526 | 35.7669 |

**Data Pre-process.** We modified the preprocessing methods based on PyHealth[9] (Yang et al., 2023a), selecting records that simultaneously contain covariates (age, weight), visit events (disease, procedure, drug), and monitoring events (lab test, injection dosage). For both tasks, we split the dataset into training, validation, and testing as 0.75: 0.1: 0.15 with the same setup of previous work (Chen et al., 2024). In the evaluation process, a bootstrapping sampling technique is employed, as in previous work (Yang et al., 2021b). The process begins with training all models on a training set, with hyperparameters selected based on a validation set. Subsequently, the evaluation is conducted by repeatedly sampling 80% of the data points from the test set with replacement. This sampling evaluation procedure is repeated over 10 rounds, and the mean and standard deviation of the results are reported as the outcomes.

## C.2 BASELINES

To validate our model, we select the following state-of-the-art methods as benchmark models for comparison.

### C.2.1 DISEASE PREDICTION

**RETAIN** (Choi et al., 2016) is an attention-based model for sequence data analysis that integrates temporal dynamics and features to predict diseases. It captures key clinical events to create patient representations.

**Transformer** (Vaswani, 2017) applies a separate transform layer for each feature and then concatenates the final hidden status of each transform layer. The concatenated hidden states are fed into the fully connected layer for prediction.

**KAME** (Ma et al., 2018) combines medical ontology knowledge to improve disease predictions. By utilizing medical knowledge, the accuracy and interpretability of predictions are improved.

**StageNet** (Gao et al., 2020) integrates a stage-aware LSTM module and a stage-adaptive convolutional module to improve predictions by considering the different stages of a patient's status.

**REFINE** (Bhoi et al., 2024) introduces monitoring-level sequences in structured EHRs, using similar temporal modelling for both visit-level and monitoring-level sequences, while incorporating personalized drug-drug interaction to capture finer-grained patient representations.

**TRANS** (Chen et al., 2024) integrates temporal edge features, global positional coding, and local structural coding into graph convolution to capture complex relationships in patient data.

---

[9]https://pyhealth.readthedocs.io/en/latest/

### C.2.2 DRUG RECOMMENDATION

**RETAIN** (Choi et al., 2016), the same approach as in disease prediction can be used for drug recommendations as well.

**Transformer** (Vaswani, 2017), the same approach as in disease prediction can be used for drug recommendations as well.

**Grasp** (Zhang et al., 2021) integrates knowledge from similar patients to enhance health representation. A single Grasp layer can be used within the model or as a standalone layer to improve other recommendation models.

**GAMENet** (Shang et al., 2019b) is based on memory networks with a memory bank enhanced by integrated drug usage, DDI (drug-drug interaction) graphs and dynamic memory with patient history.

**SafeDrug** (Yang et al., 2021b) introduces drug-related molecular knowledge and learns drug interactions through molecular characterization to recommend safer drug combinations.

**Micron** (Yang et al., 2021a) uses a recurrent residual learning model to predict medication changes, then recommends based on those changes and the previous visit's drug combination.

**MoleRec** (Yang et al., 2023b) delves into the importance of specific molecular substructures in drugs. This approach enhances the accuracy of drug recommendations by leveraging finer molecular representations.

**REFINE** (Bhoi et al., 2024), the same approach as in disease prediction can be used for drug recommendations as well.

**TRANS** (Chen et al., 2024), the same approach as in disease prediction can be used for drug recommendations as well.

**CausalMed** (Li et al., 2024) utilizes causal discovery based on patient status to identify primary and secondary diseases, thereby enhancing personalized patient representation.

### C.3 EVALUATION METRICS

Our task scenario is based on EHR data mining within the clinical medical system, while the multi-label prediction is part of the recommender system domain. Therefore, we employ two sets of evaluation metrics to evaluate our work. The following description uses drug recommendation as an example, and the same evaluation metrics apply to disease prediction.

### C.3.1 MEDICAL SYSTEM

From the perspective of the medical system, we use four main general metrics (according to Jiang et al. (2023)) to evaluate the performance of our method: the F1-score, Jaccard, PR-AUC, and ROC-AUC.

**F1-score** combines precision and recall, reflecting the model's ability to accurately identify correct drugs while ensuring comprehensive coverage.

$$\text{Precision}(t) = \frac{|\{i : \hat{d}_i = 1\} \cap \{i : d_i = 1\}|}{|\{i : \hat{d}_i = 1\}|}, \tag{10}$$

$$\text{Recall}(t) = \frac{|\{i : \hat{d}_i = 1\} \cap \{i : d_i = 1\}|}{|\{i : d_i = 1\}|}, \tag{11}$$

$$\text{F1}(t) = \frac{2}{\frac{1}{\text{Precision}(t)} + \frac{1}{\text{Recall}(t)}}, \tag{12}$$

$$\text{F1} = \frac{1}{T_H} \sum_{t=1}^{T_H} \text{F1}(t), \tag{13}$$

where $\hat{d}_i$ represents the predicted outcome, $d_i$ represents the real label, $T_h$ represents the total number of visits for patient $H$.

**Jaccard** is employed to evaluate the similarity between two sets. In drug recommendation, a higher Jaccard score indicates that the predicted prescription is more consistent with the actual drug regimen, indicating higher accuracy.

$$\text{Jaccard}(t) = \frac{|\{i : \hat{d}_i = 1\}| \cap |\{i : d_i = 1\}|}{|\{i : \hat{d}_i = 1\}| \cup |\{i : d_i = 1\}|}, \tag{14}$$

$$\text{Jaccard} = \frac{1}{T_H} \sum_{t=1}^{T_H} \text{Jaccard}(t). \tag{15}$$

**PR-AUC** assesses model performance across different recall levels, indicating the ability to maintain precision with increasing recall.

$$\text{PR-AUC}_t = \sum_{k=1}^{|D|} \text{Precision}_{k_t} \triangle \text{Recall}_{k_t}, \tag{16}$$

$$\triangle Recall_{k_t} = \text{Recall}_{k_t} - \text{Recall}_{k-1_t}, \tag{17}$$

where $|D|$ denotes the number of drugs, $k$ is the rank in the sequence of the retrieved drugs, and $Precision_k(t)$ represents the precision at cut-of $k$ in the ordered retrieval list and $\triangle Recall_{k_t}$ denotes the change in recall of a drug's ranking from $k-1$ to $k$. We averaged the PR-AUC across all of the patient's visits.

**ROC-AUC** calculates the area under the ROC curve by summing the areas of trapezoids formed between consecutive points on the ROC curve.

$$\text{ROC-AUC} = \sum_{i=1}^{n-1} \frac{1}{2} \times (FPR_{i+1} - FPR_i) \times (TPR_{i+1} + TPR_i), \tag{18}$$

where $FPR_i$ and $TPR_i$ are the false positive rate and the true positive rate at the $i$ threshold, respectively, and $n$ is the number of thresholds.

### C.3.2 RECOMMENDER SYSTEM

From a recommender system perspective, we use the visit-level precision@k and event-level accuracy@k (according to Chen et al. (2024)) to evaluate our methods.

**Visit-level Precision@k** measures the precision of individual visit. Visit-level precision@k is defined as the number of correct visit events in the top-ranked $k$ predictions divided by $\min(k, |D_t|)$, where $|D_t|$ is the number of category labels of target events in visit $v_t$. We report the average visit precision@k for all visits. The visit-level precision @k is defined as:

$$\text{visit-level precision@}k = \frac{\sum_{i=1}^{k} \text{I}(\hat{d}_i = d_i)}{\min(k, |D_t|)}, \tag{19}$$

where the numerator represents the number of correct predictions in the top-k prediction, which are ordered by probability.

**Event-level Accuracy@k** measures the overall accuracy of the model's predictions and is defined as the number of correctly predicted visit events divided by the total number of top-ranked $k$ predicted visit events. For multiple visit sequences, the event-level accuracy @k is defined as:

$$\text{event-level accuracy@}k = \frac{\sum_{t=1}^{|V|} \sum_{i=1}^{k} \text{I}(\hat{d}_i = d_i)}{\sum_{t=1}^{|V|} |D_t|}, \tag{20}$$

where $|V|$ denotes the total number of visits.

The average number of diseases per visit is between 10-20, while the average number of drugs per visit is between 30-40, so we set $k$ to 10, 20, 30 in disease prediction and set 30, 40, 50 in drug recommendation to evaluate the coarse-grained and fine-grained performance of each model.

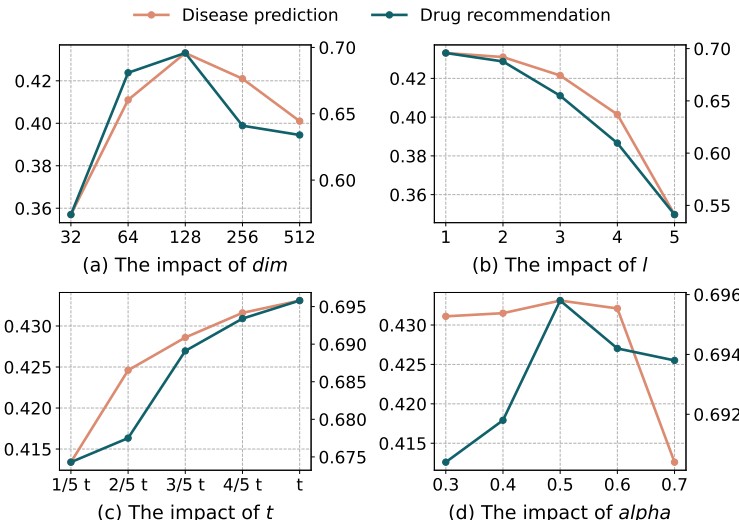

Figure 11: Hyperparameter testing, represented by the F1-score on the MIMIC-III dataset.

### C.4 IMPLEMENTATION DETAILS

**Experimental Environment** The experiments are carried out on an Ubuntu 22.04 system equipped with 80GB of memory, a 32-core CPU, and a 48GB A40 GPU, utilizing Python 3.8.16, PyTorch 2.0.0 and CUDA 11.7.

## D ADDITIONAL EXPERIMENTS

### D.1 PARAMETER SENSITIVITY STUDY

To achieve optimal model performance and analyze the impact of key parameters, we conduct hyperparameter tests in this subsection. We evaluate the effects of Embedding dimension ($dim$), Number of graph layers ($l$), Number of graph propagation steps ($t$), and Residual ratio ($\alpha$). Figure 11 shows the model's F1-score performance on drug recommendation and disease prediction tasks using the MIMIC-III dataset under different parameter settings.

**Embedding dimension** $dim$**.** To evaluate the effect of the embedding dimension on the performance of the proposed model, we conduct scaling experiments on the size of the embedding dimension, and the results are shown in Figure 11 (a). It is found that the performance of CrossMed improves significantly as the embedding dimension increases and reaches an optimum at a dimension of 128. However, an embedding dimension beyond 128 leads to a gradual decrease in performance, a trend that occurs simultaneously in both tasks. The performance degradation may be due to the introduction of noise by too large a dimension, which in turn leads to overfitting of the model. Based on the experimental results, we finally chose to set the embedding dimension to 128.

**Number of graph layers** $l$**.** In our model, the number of graph network layers is crucial for capturing the pathological relationships between medical events and personal information in cross-level feature interaction. Figure 11 (b) illustrates the experimental results, showing that the model achieves optimal performance when the number of graph network layers is 1. This is because the heterogeneous network structure proposed in this paper is relatively complex and saturated at a layer number of 1. Further increasing the layer number not only fails to improve performance, but also may lead to overfitting and reducing generalization ability.

**Number of graphs propagation times** $t$**.** As shown in Figure 11 (c), the two tasks exhibit the same trend: as the number of graph propagation increases, the accuracy of the computational results significantly improves. In the case where the total length of the sequence is $t$, the results show a significant improvement when the number of propagation is increased from 1/5 $t$ to 3/5 $t$, while the results are still improved but to a lesser extent when the number of propagation times is increased

Table 6: Difference in results between independent and combined training

| Model | Disease prediction | | | | Drug recommendation | | | |
|---|---|---|---|---|---|---|---|---|
| | F1-score | Jaccard | PR-AUC | ROC-AUC | F1-score | Jaccard | PR-AUC | ROC-AUC |
| Independent | 43.33 (1.4) | 27.80 (1.1) | 51.37 (1.4) | 92.43 (0.4) | 69.58 (2.3) | 53.35 (2.7) | 79.02 (1.8) | 93.61 (0.6) |
| Combined | 40.18 (1.2) | 22.80 (1.3) | 49.92 (1.5) | 91.45 (0.7) | 66.48 (1.0) | 49.62 (2.4) | 77.21 (1.5) | 91.37 (0.7) |

from 3/5 $t$ to $t$. The results suggest that the higher the number of propagation times, the more information about the early time points is absorbed. Meanwhile, data near the end of the sequence have a greater impact on the results, and although the influence of early data is not as significant as later data, it still provides a significant gain.

**Residual ratio** $\alpha$. $\alpha$ controls the proportion between a node's representation before and after integrating additional information. Specifically, a larger $\alpha$ indicates a greater focus on the updated representation, while a smaller $\alpha$ emphasizes the representation prior to updating. As shown in Fig. 11(d), both tasks achieve optimal performance when $\alpha$ equals 0.5. If $\alpha$ is either too small or too large, performance declines.

### D.2 MULTI-TASK TRAINING

The proposed CrossMed is a general model, and through the aforementioned experiments and analyses, we have demonstrated its capability to be applied independently to clinical diagnosis and treatment tasks. To further investigate its generalization ability—specifically, whether it can be trained once to perform well across multiple tasks—we conducted the following multi-task training experiments.

**Independent training**: each task is looked at using the same model but with a specialized set of parameters, i.e., for each task, we train a model from scratch and optimize its parameters for that task.

**Combined training**: each task is looked at with the same model and the same parameters, i.e., multiple loss functions are merged, the model is trained on multiple tasks at the same time, and the parameters from one training are directly applied to multiple tasks.

Table 6 demonstrates the performance of independent training versus combined training in the disease prediction and drug recommendation tasks in the medical system evaluation under the MIMIC-III dataset. The results show a significant decrease in accuracy for both tasks. Combined training may introduce information that is irrelevant to the task at hand. CrossMed incorporates cross-level feature interaction modules, allowing it to efficiently filter out irrelevant information for independent tasks using an explicit objective loss function. On the contrary, when the model needs to process multiple tasks simultaneously, it may introduce information that is favourable to one task but unfavourable to other tasks, thus affecting the overall performance. In summary, the combined training approach fails to optimize for a specific task, resulting in a degradation of model performance on certain tasks.

