# OpenReview forum: "Temporal Visiting-Monitoring Feature Interaction Learning for Modelling Structured Electronic Health Records"
_ICLR.cc/2025/Conference — Submitted to ICLR 2025_

### Official Review · Reviewer_MW7W · 2024-11-02

**Soundness:** 1
**Presentation:** 4
**Contribution:** 3
**Rating:** 3
**Confidence:** 4

**Summary:**

This paper seeks to improve the ability to make predictions on electronic health record data by constructing a graph between features and then using a graph neural network. The key insight is that we can split EHR data features into two categories, monitoring events (where we measure something about the patient) and visit events (where we do something about those recordings) and carefully construct the graph to take advantage of the relationship between the two. The result is improved predictive performance.

**Strengths:**

-  Well written paper, with very nice text, figures and appendix
-  Good evaluation task and metric selection

**Weaknesses:**

- There are very minimal details on how hyperparameter selection was done for this method and the baselines. (I don’t see any details at all for the baselines?) Ideally we want a table of the optimal hyperparameters and the hyperparameter search grid (or if random search, the distributions searched) for every model, including your proposed model and baselines. Did you dedicate roughly equal time to hyperparmeter searching for both your method and your baselines?

- Almost all (except 2) of the baselines are missing monitoring event features, which is a very significant limitation. I think it’s necessary to have at least one simple baseline that has both monitoring events and visit events as features. Based on the fact that most of your sequences have length 1 (and your experiments that show that sequence length doesn't really matter), a very simple feedforward neural network that has access to all features would be a great baseline.

- These are very small datasets, 15k for MIMIC-III and 19k for MIMIC-IV. I am skeptical that this method will continue to provide benefits for larger datasets, as in theory with enough data a deep neural network should be able to automatically learn these interactions. This ties into how even the authors note that the dataset is so small that a single graph layer performs best.

- A major part of the data processing code “preprocess.drug_recommendation_mimic34_fn” and “preprocess.diag_prediction_mimic34_fn” is missing. This will make this paper more challenging to reproduce and makes it difficult for me to fully review / understand this paper.

- The improvements in performance are small, and (if I am reading table 1 correctly) are often within a standard deviation of the baselines.

**Questions:**

See weaknesses

---

> ### Author Response · Authors · 2024-12-01
>
> Thank you for your feedback on our work. Below, we have provided detailed responses to your questions. A revised version has been uploaded with the changes clearly highlighted in red.
>
> **[W1] Did you dedicate roughly equal time to hyperparmeter searching for both your method and your baselines?**
>
> >Thank you for your question. To ensure fairness in our experiments, we followed to the approach outlined in the pyhealth project [1], which is a Python library designed for clincal prediction tasks on EHR data. This library provides data preprocessing methods, classical baselines, and their corresponding optimal parameters. For the baselines included in the library (such as RETAIN, Transformer, KAME, StageNet, Grasp, GAMENet, SafeDrug, Micron, MoleRec), we directly used the optimal parameters offered in the library. For the baslines not included in the library (such as REFINE, Trans, CausalMed, and Ours), we either selected the optimal parameters provided in the original papers or conducted additional hyperparameter tuning.
> >
> > [1] Yang C, Wu Z, Jiang P, et al. PyHealth: A deep learning toolkit for healthcare predictive modeling[C]//Proceedings of the 27th ACM SIGKDD International Conference on Knowledge Discovery and Data Mining (KDD). 2023, 2023.
>
> **[W2] I think it’s necessary to have at least one simple baseline that has both monitoring events and visit events as features.**
>
> > Thank you for your suggestion. We have included a simple baseline using MLP and integrated it into the performance comparison section of the manuscript. The modifications in the manuscript are marked in red text.
>
> | Model  | MIMIC-III | |  | | MIMIC-IV|  |  | |
> |-|-|-|-|--------|----------------|----------------|----------------|----------------|
> |  | F1-score       | Jaccard        | PR-AUC         | ROC-AUC        | F1-score       | Jaccard        | PR-AUC         | ROC-AUC        |
> | Disease Prediction
> | MLP                   | 37.21 (1.0)    | 23.17 (0.8)    | 42.13 (1.8)    | 89.93 (0.4)    | 40.25 (0.8)    | 24.78 (0.7)    | 45.43 (1.6)    | 90.74 (0.4)    |
> | REFINE                | 38.84 (1.9)    | 24.10 (1.4)    | 49.53 (1.2)    | 92.09 (0.4)    | 41.14 (1.1)    | 25.90 (0.9)    | 51.80 (1.3)    | 92.66 (0.2)    |
> | Ours                  | 43.31 (1.4)*   | 27.80 (1.1)*   | 51.37 (1.4)*   | 93.43 (0.4)*   | 46.82 (1.7)*   | 29.78 (0.9)*   | 52.16 (1.3)    | 93.10 (0.6)*   |
> | Drug Recommendation
> | MLP                   | 63.64 (2.1)    | 46.67 (2.3)    | 69.72 (2.3)    | 89.38 (0.9)    | 62.75 (2.2)    | 47.87 (2.4)    | 67.88 (4.5)    | 90.37 (1.2)    |
> | REFINE                | 66.73 (2.6)    | 50.07 (2.9)    | 78.25 (2.6)    | 92.54 (0.7)    | 66.05 (1.0)    | 49.66 (1.1)    | 78.29 (1.5)    | 92.11 (0.3)    |
> | Ours                  | 69.58 (2.3)    | 53.35 (2.7)    | 79.02 (1.8)    | 93.61 (0.6)    | 68.74 (2.5)*   | 52.37 (2.8)    | 79.34 (2.5)    | 94.12 (1.7)*   |
>
> **[W3] These are very small datasets, I am skeptical that this method will continue to provide benefits for larger datasets, as in theory with enough data a deep neural network should be able to automatically learn these interactions.**
>
> > Thank you for your comments.
> > (1) We fully agree that with access to billions of data points and sufficiently large models, these interactions could be learned automatically without the need for manual definitions.
> > (2) However, due to the privacy and intellectual property constraints of medical data, collecting such datasets is extremely challenging, and datasets containing detailed monitoring session information are even rarer.
> > (3) In fact, the MIMIC-III and MIMIC-IV datasets are currently the largest publicly available and widely recognized medical databases globally. Nevertheless, their size is still relatively small compared to datasets in other fields.
> > (4) Our proposed method aims to maximize feature interactions on limited datasets to enhance model performance for clinical prediction tasks.
>
> **[W4] A major part of the data processing code is missing. This will make this paper more challenging to reproduce and makes it difficult for me to fully review / understand this paper.**
>
> > Thank you for your question.
> > (1) The data processing method is indeed relatively complex and incurs significant time overhead.
> > (2) To address this, we have submitted some preprocessed data samples, such as "CrossMed/data/mimic3/processed_data/drug_rec/processed_developer_data.pkl", so that anyone can directly run and understand the experiments described in the paper.
> > (3) To ensure transparency, we will release all code on GitHub after the publication of our work.
>
> **[W5] The improvements in performance are small, and (if I am reading table 1 correctly) are often within a standard deviation of the baselines.**
>
> > Thank you for your question. We will report the t-test results, p-values, and their significance for each experimental result, and update the manuscript accordingly. The results are the same as the table shown in [W2].

---

> ### Comment · Reviewer_MW7W · 2024-12-01
> **Quick clarification questions**
>
> Thank you for the detailed responses, which I am still reading though. But I had some quick clarification questions that I wanted to ask first (
> due to the short response period )
>
> 1. Precisely how are those p values being calculated? You are doing a t test between what and what, using what n, etc.
>
> 2. It looks like you will not be releasing either the hyperparemter search grid or your data processing code during this review process?

---

### Official Review · Reviewer_zWVb · 2024-11-03

**Soundness:** 3
**Presentation:** 2
**Contribution:** 2
**Rating:** 3
**Confidence:** 4

**Summary:**

This paper introduces CrossMed, a method for modeling EHRs that enhances disease prediction and drug recommendation by integrating both visit-level and monitoring-level data, and they claim incorporating finer-grained monitoring sequences simultaneously in structured EHRs, where each visit involves multiple monitoring sessions, can improve prediction performance. CrossMed’s cross-level feature interaction approach captures dynamic patient health trends. it is tested on MIMIC-III and MIMIC-IV datasets for both diagnosis prediction and drug recommendations.

**Strengths:**

1. One of the few papers focusing on combining visit-level EHR discrete code sequence data with continuous monitoring measurements, providing a finer-grained view of a patient's medical information.
2. Comprehensive experimentation on both diagnosis prediction and drug recommendation tasks.

**Weaknesses:**

1. The explanation of "monitoring data" is unclear. It mentions “monitoring-level events, such as lab test results reflecting the patient’s health state.” Are these continuous signals like ECG (time series), imaging data like MRI, or simply numerical lab test results? A clearer definition is needed.
2. The methodology formulation could be enhanced with a detailed figure to improve understanding. Current figures only depict the high-level concept. Figure captions require more detail to be self-explanatory (e.g., Figure 3).
3. If monitoring data includes lab signals and measurements, this would classify the work as multimodal. Comparisons should then be made with works using similar data modalities. However, the baselines used here only involve EHR data (sequences of discrete codes, such as diagnoses, prescriptions, and procedures, inside hospital visits).

**Questions:**

Please clearly explain monitoring data with examples. Do all the patients contains these information (Provide statistics)?

---

> ### Author Response · Authors · 2024-12-01
>
> Thank you for your valuable feedback on our work. Below, we provide detailed responses to your questions. A revised version has been uploaded, with the new changes clearly marked in red.
>
> **[W1] The explanation of "monitoring data" is unclear. A clearer definition is needed.**
>
> > Thank you for pointing out the issue.  In this paper, monitoring data includes only quantified lab test results and injection dosage information. In addition, in line 714 of Appendix A.1, we provided an explanation of what constitutes monitoring data.
>
> **[W2] Current figures only depict the high-level concept. Figure captions require more detail to be self-explanatory (e.g., Figure 3).**
>
> >Thank you for your suggestion. We revised the caption to include more detailed and specific information, ensuring a more comprehensive and thorough description. The modifications in the manuscript are marked in red text.
>
> **[W3] If the monitoring data includes lab signals and measurements, the work qualifies as multimodal.**
>
> >Thank you for your insightful comment. Indeed, lab tests are also a part of EHR data, and their data format is similar to diagnoses, prescriptions, and procedures, as they are represented as sequences of discrete codes. The only distinction is that lab tests pertain to more fine-grained events. While sequences of diagnoses are continuous across visits, sequences of lab tests are continuous within monitoring sessions. Therefore, they do not constitute a different modality.
>
> **[W4] Please clearly explain monitoring data with examples. Do all the patients contains these information (Provide statistics)?**
>
> >Thank you for your question. The data processing method used in our work is consistent with previous studies [1]. In those studies, samples with records of disease, procedure, and drug (visit events) were selected. Based on this, we further filtered samples which also had lab test results and injection records (monitoring events).
> >
> >In the MIMIC-III dataset: The number of samples meeting the criteria of previous studies is 42,457. Among them, 18,557 samples have monitoring data, accounting for 43.70%.
> >
> >In the MIMIC-IV dataset: The number of samples meeting the criteria of previous studies is 173,845. Among them, 24,777 samples have monitoring data, accounting for 14.25%.
> >
> >[1] Yang C, Wu Z, Jiang P, et al. PyHealth: A deep learning toolkit for healthcare predictive modeling[C]//Proceedings of the 27th ACM SIGKDD International Conference on Knowledge Discovery and Data Mining (KDD). 2023, 2023.

---

### Official Review · Reviewer_1UkH · 2024-11-03

**Soundness:** 3
**Presentation:** 3
**Contribution:** 2
**Rating:** 5
**Confidence:** 5

**Summary:**

In this work, the authors propose an EHR modeling method focusing on utilizing multi-level interactions. The experiments are conducted using MIMIC-III and IV datasets with two tasks.

**Strengths:**

The paper is well-written, and the experiments are comprehensive.

**Weaknesses:**

My major comments are:

1. The current methodology contribution and insights are somewhat incremental. Multi-level EHR data modeling is not a new topic; for example, [1] and some of its subsequent works are not discussed in this paper.
[1] MiME: Multilevel Medical Embedding of Electronic Health Records for Predictive Healthcare

2. What are the causal assumptions in this work? Why are there only responses from visit to monitoring events? What outcomes are used as targets, and what features are used as confounders? How are such causal assumptions proposed?

3. Can the dataset reflect the insights? MIMIC-III and MIMIC-IV are ICU datasets. The monitoring events are within each ICU stay. Do the monitoring events in one ICU stay affect the next visit?

4. Some experiment results are very close. P-values should be reported.

5. The major contribution of this work is about cross-level interactions. However, the authors did not provide any analysis of the interaction results. The information in the t-SNE plot is not straightforward. Why does sparser indicate better results? I would assume forming subclusters is more useful in practice. Besides, the difference in the patient representation plot looks the same to me.

**Questions:**

Please refer to weaknesses

---

> ### Author Response · Authors · 2024-12-01
>
> Thank you for recognizing the strengths of our work. Below are detailed responses to your questions. We have uploaded a revised version with the new changes highlighted in red.
>
> **[W1] Multi-level EHR data modeling is not a new topic; for example,  MiME and some of its subsequent works are not discussed in this paper.**
>
> > Thank you for highlighting this issue. The "structured EHR data" referred to in MiME and most prior works differ from the "structured EHR data" discussed in this paper.
> > In MiME, a single visit session consists of multiple visit events (e.g., disease, procedure, drug). In contrast, our structured EHR data includes not only visit events within a visit session but also monitoring sessions, where each monitoring session comprises multiple monitoring events (e.g., lab test, injection).
> > To put it simply, the “multi-level” concept in previous works refers to the hierarchy between folders and files (i.e., sessions and events). In contrast, our approach focuses on the hierarchy between folders and folders (i.e., sessions and sessions).
>
> **[W2] (1) What are the causal assumptions in this work? (2)Why are there only responses from visit to monitoring events? (3)What outcomes are used as targets, and what features are used as confounders? How are such causal assumptions proposed?**
>
> >Thank you for your insightful questions.
> >
> > (1)This study primarily focuses on the correlations between multiple events and does not involve causal effect estimation or causal discovery. Futhermore, we plan to extend this method to incorporate causal assumption in future work, which could provide a more comprehensive perspective.
> >
> > (2)Within a single visit, multiple monitoring sessions are included. At the same time point, we model the relationship from monitoring events to visiting events, while across different time points, the relationship from visiting events to monitoring events is also modeled. This results in a bidirectional relationship, capturing not only the response from visiting events to monitoring events but also the influence of monitoring events on visiting events.
> >
> > (3)This work does not involve any causal assumptions. All variables, such as laboratory results, are treated as covariates, and the outcomes like predicted diseases or drugs are treated as targets.
>
> **[W3] The monitoring events are within each ICU stay. Do the monitoring events in one ICU stay affect the next visit?**
>
> >Thank you for your question. Our method does not involve the monitoring data from the first visit influencing the second visit. As stated in line 233 of the paper: "Since diseases are visit-level events, all  within the same visit  are initialized to be identical." We consider that within a single visit session, the evolving monitoring data continuously impacts the current visit as a whole, rather than influencing the next visit.
>
> **[W4] Some experiment results are very close. P-values should be reported.**
>
> >Thank you for your suggestion. We will report the t-test results, P-values, and their significance for each experimental result, and update the manuscript accordingly. The modifications in the manuscript are marked in red text.
>
> | Task               | MIMIC-III  |        |        |        | MIMIC-IV   |        |        |        |
> |--------------------|------------|--------|--------|--------|------------|--------|--------|--------|
> |                    | F1-score   | Jaccard| PR-AUC | ROC-AUC| F1-score   | Jaccard| PR-AUC | ROC-AUC|
> | Disease Prediction | 43.31(1.4)*| 27.80(1.1)*| 51.37(1.4)*| 93.43(0.4)*| 46.82(1.7)*| 29.78(0.9)*| 52.16(1.3)| 93.10(0.6)*|
> | Drug Recommendation| 69.58(2.3) | 53.35(2.7)| 79.02(1.8) | 93.61(0.6) | 68.74(2.5)*| 52.37(2.8)| 79.34(2.5)| 94.12(1.7)*|
>
> **[W5] The authors did not provide any analysis of the interaction results. The information in the t-SNE plot is not straightforward. Why does sparser indicate better results? I would assume forming subclusters is more useful in practice.**
>
> > Thank you for pointing out this issue. Both Fig. 7 and Fig. 8 provide analyses of cross-level interactions:
> >
> > (1) Regarding Fig. 7: The results demonstrate that cross-level interaction outperforms parallel interactions and one-directional interaction. We also provide an analysis explaining why this is the case.
> >
> > (2) Regarding Fig. 8: For the t-SNE visualization, you mentioned that closer node proximity indicates better performance. This holds true for standard multi-class classification tasks, where tighter clusters often imply better separability. However, our task is a multi-label binary classification problem, aiming to extract a set of medications from 200 possible drugs. This effectively transforms the problem into a multi-class classification task with  potential classes. In this context, overly close clustering would undermine the individualization of patient-specific treatments.

---

### Official Review · Reviewer_Gime · 2024-11-04

**Soundness:** 3
**Presentation:** 2
**Contribution:** 3
**Rating:** 5
**Confidence:** 4

**Summary:**

This paper presents a method (CrossMed) to model visit-level and finer-grained monitoring-level EHR data simultaneously.
The method first builds a cross-level interaction temporal graph for each visit at monitoring-level.
A temporal network then aggregates the temporal graph into a visit embedding for further downstream tasks.
Evaluation on disease prediction and drug recommendation tasks shows that CrossMed outperforms current state-of-the-art models.

**Strengths:**

The proposed approach is able to integrate both visit-level and monitoring-level data in EHRs, which allows it to capture finer-grained health trends.

The authors demonstrate the effectiveness of the proposed modules through several ablation studies.

**Weaknesses:**

The size of the temporal graph could become large if there are many monitoring sessions during a single visit, which would significantly increase the computational cost.

Some parts of the proposed model are not well-introduced or the explanation is unclear, making this paper somewhat hard to follow.
For example, the visit-to-patient stage is also not well explained in the main text.
It is unclear how the patient representation and visit representation are combined.
These details are important to understand how the authors derive the final patient representation $h_H$.
I would suggest including the content in Appendix B.4 in the main text and elaborating on it in more detail.

**Questions:**

It seems that in the cross-level interaction temporal graph, all monitoring event-disease pairs are the same, represented by the aggregate value generated from the relationship discovery module.
Why not use the unaggregated values to give different monitoring event-disease pairs different weights? (same question for m-P and m-R pairs)

How do the authors choose $\alpha$ as mentioned in Eq 3? Is $z_i$ in Eq 4 a trainable parameter, or is it a fixed value?

What does the "temporal recurrent component in feature interaction" refer to in the ablation study? Does it refer to the GRU structure used to aggregate $h_vt$ or the feature interaction on cross-time edges.

---

> ### Author Response · Authors · 2024-12-01
>
> Thank you for your thoughtful feedback on our work. Below, we provide detailed responses to your questions. A revised version has been uploaded with the changes clearly highlighted in red.
>
> **[W1] A large number of monitoring sessions per visit could inflate the temporal graph size, increasing computational costs.**
>
> >Thank you for your insightful comment. We acknowledge that our approach incurs higher time and computational costs during the training phase compared to other methods, as we decompose a single visit into multiple monitoring sessions. However, this additional cost is incurred only once during training. Given the significant improvements in predictive performance, we believe this overhead is negligible. In practical applications, during the inference phase, the time required to predict a disease or prescription for a single patient on a CPU is approximately 0.2 seconds.
>
> **[W2] Some parts of the model are poorly explained, making the paper hard to follow.**
>
> > Thank you for pointing out the problem. We have incorporated B.4 into the main text and made the necessary revisions.
> >
> > Each cross-level temporal graph captures the representation of the last time point within its respective sequence.  (1) In the monitoring-to-visit stage, we derive the representations of the last monitoring session and concatenate these representations to obtain the visit representation,  $h_{V_t}$ .  (2) Similarly, in the visit-to-patient stage, we derive the representations of the last visit and concatenate them into the patient representation,  $h_{H}$ , which serves as the final representation for downstream tasks.
>
> **[Q1]  Why not use unaggregated values to assign different weights to each monitoring event-disease pair?**
>
> >Thank for your excellent question. Initially, we considered establishing edges between each monitoring event and visit event based on computed weights. This approach, in practical experiments, would significantly increase the complexity of the graph structure, resulting in a sharp rise in the number of feature interactions and, consequently, a substantial increase in computational cost. However, we found that this method did not yield a significant improvement in performance. Therefore, in balancing efficiency and effectiveness, we opted to use aggregated weights for the edges.
>
> **[Q2] How do the authors choose α as mentioned in Eq 3? Is $z_i$ in Eq 4 a trainable parameter, or is it a fixed value?**
>
> >Thank you for your question. Regarding your questions: (1) Selection of α in Equation 3: α is a hyperparameter that we determined through hyperparameter tuning. We evaluated various α values on a validation set and selected the one that optimized the model’s performance, as shown in the Fig.11 (d) in the Appendix D.1. (2) Nature of zᵢ in Equation 4: The variable zᵢ in Equation 4 is a trainable parameter. It is initialized randomly and updated during the training process through backpropagation to minimize the loss function.
>
> **[Q3] What does the “temporal recurrent component in feature interaction” refer to in the ablation study?**
> > Thank you for your question. The statement refers to “the feature interaction on cross-time edges.”

---

### Meta-Review · Area_Chair_ZzkY · 2024-12-20

**Metareview:**

The paper introduces an EHR modeling method CrossMed, which models the dynamic interaction between visit-level and monitoring-level data and captures finer-grained health trends. CrossMed first captures the dynamic influence between medical data and then performs a visiting-monitoring feature interaction on the relationships between visit data and monitoring data. Experiments of disease prediction and drug recommendation tasks on publicly available MIMIC-III and MIMIC-IV datasets show the good performance of CrossMed. While the topic is healthcare-related, both tasks and data are standard and not disease-specified, which is far from real clinical decision-making. After the rebuttal stage, reviewers still kept some unsolved concerns, such as the novelty of the framework from an ML perspective, the causal assumption in problem formulation, small ICU data being different from real problems, and marginal performance improvement.

**Additional Comments On Reviewer Discussion:**

There are some discussions between the reviewers and authors. The authors have not completely addressed all the questions raised by reviewers. All reviewers maintain their scores. During the discussion phase, no reviewer championed the paper.

---

### Decision · Program_Chairs · 2025-01-22

Reject